# Well-Being of Family Caregivers of Individuals with Spinal Cord Injury: The Moderating Effects of Online Versus In-Person Social Support

**DOI:** 10.3390/ijerph22071075

**Published:** 2025-07-05

**Authors:** Victoria Bogle, William C. Miller, Heather Cathcart, Somayyeh Mohammadi

**Affiliations:** 1Department of Psychology, Kingston University, London KT1 2EE, UK; victoriacc6@gmail.com; 2Department of Occupational Science and Occupational Therapy, University of British Columbia, Vancouver, BC V6T 2B5, Canada; cathcart.heather@gmail.com (H.C.); somayyeh.mohammadi@ubc.ca (S.M.); 3Rehabilitation Research Program, GF Strong Rehabilitation Centre, Vancouver, BC V5Z 2G9, Canada; 4Centre for Aging SMART, Vancouver Coastal Health Research Institute, Vancouver, BC V6H 3Z6, Canada

**Keywords:** social support, family caregivers, burden, SCI, social media

## Abstract

Objective: Family members of individuals with spinal cord injury often take on caregiving responsibilities, which can lead to caregiver burden. One factor that can mitigate the adverse effects of caregiving, such as caregiver burden, is receiving social support. Caregivers can obtain support from people they meet in person (in-person support) and on social media platforms (online support). The current cross-sectional correlational design study investigated the moderating effect of in-person and online support on the association between relationship quality, caregiver competence, caregiver distress, and caregiver burden (dependent variables). Methods: Family caregivers of an individual with spinal cord injury (*n* = 115) completed an online survey assessing relationship quality, competence, distress, burden, and in-person and online supports. Results: Moderation analyses showed that the negative associations between relationship quality and physical burden (B = −0.58; *p* = 0.019) and caregiver competence and physical burden (B = −0.73; *p* = 0.013) were more pronounced at higher levels of online social support. Furthermore, the magnitude of the negative associations between relationship quality and emotional burden (B = −0.52; *p* < 0.001) and caregiver competence and emotional burden (B = −0.34, *p* = 0.012) were more pronounced at higher levels of in-person social support. Moderation analyses also revealed that the positive association between distress and social burden (B = 0.47; *p* = 0.029) and emotional burden (B = 0.26; *p* = 0.045) were stronger when caregivers reported higher levels of online support. Conclusions: In-person and online support can buffer some aspects of caregiver burden on caregiver well-being. While online support is usually considered beneficial, greater online engagement may contribute to higher levels of burden when the distress is high. It is possible, however, that caregivers who are more distressed engage more with online media to receive support.

## 1. Introduction

Spinal cord injury (SCI) is one of the leading causes of paralysis. Most individuals with SCI rely on their family members, referred to as family caregivers, for assistance with daily life activities such as bathing and meal preparation [1]. The economic value of caregivers’ unpaid care has been estimated at around USD 600 billion in the United States alone [2]. However, caregiving responsibility may lead to caregiver burden, defined as the impact of caregiving on social, emotional, and physical well-being [3]. The well-established research in the caregiving field shows that depression, anxiety, and lower physical health are commonly reported by caregivers [4]. Specific to the SCI community, data has shown that caregivers of individuals with SCI are more likely to experience heart disease (12% vs. 6%), obesity (43% vs. 28%), physical distress (20% vs. 12%), and insufficient sleep (47% vs. 30%) in comparison to caregivers of individuals with neurological diseases [5].

The adapted stress model [6] suggests that general stressors related to caregiving, such as care demands and problematic behaviours (e.g., agitation and aggression), influence the primary assessment of the caregiving role and are directly linked to caregiver burden. Consequently, caregivers conduct a secondary assessment to evaluate their resources for managing the stressors, which can lead to role and intrapsychic stressors, such as conflicts among the various roles caregivers play and changes in self-competence, and subsequently, increase levels of distress and burden [6]. The adapted stress model also explains that caregiving influences the quality of the relationship between the caregiver and the individual with SCI [6]. For example, protective behaviours such as hiding their worries can lead to communication breakdown and relationship deterioration [7]. In addition, the responsibility placed on the caregivers can cause resentment towards the individual with SCI; thus, straining the relationship, especially if caregivers feel unappreciated [1]. Reduced relationship quality can contribute to caregiver burden [8].

Emotional distress, generated from role overload, is another leading contributor to the burden [9], causing caregivers to feel overwhelmed and negatively impacting their ability to provide care [10]. These feelings may influence caregivers’ appraisal of situations, leading them to perceive their role negatively, contributing to an even higher burden [11]. Negative emotions can be linked to low levels of perceived competence [6,7]. Research has shown that increased perceived competence in caregivers can improve social functioning and mental and physical health, reducing the burden [1].

Based on the adapted stress model, social support can protect against adverse caregiving outcomes [12]. Peer support groups have effectively reduced the burden by helping caregivers adapt to their roles. These groups can reduce feelings of isolation, maintaining and improving the relationship between caregivers and individuals with SCI [7]. Similarly, caregivers with access to community services that provide support and training, such as problem-solving and conflict-resolution skills, report feeling more competent in their role and experiencing less burden [1]. Those who receive social support may have a positive caregiving experience, which helps them maintain or increase their emotional reserve. Expanding this reserve can alleviate distress and burden [6]. Social support may also facilitate healthy behaviours, diminish the impact of stressors, and reduce their burden [13].

While studies have highlighted the importance of social support for caregivers to reduce the burden [14,15], some caregivers cannot access in-person social support due to financial difficulties or a lack of time to travel to support networks [16]. Therefore, an increasing number of caregivers are turning to online social network services, such as Facebook and Instagram [17,18], to overcome accessibility barriers. Those using online social networks to receive social support join a community of caregivers to whom they can relate and access emotional, companionship, appraisal, and informational support [11,18]. Support received via online platforms offers unique features that cannot be replaced by in-person social support [19]. For example, online social support allows caregivers to be anonymous, which may help them feel more comfortable sharing negative emotions [16].

While some studies point to the benefits of in-person and online support for caregivers [14], the moderating effects of this support, especially online social support, on caregiver burden have not been investigated. Therefore, this study assesses the moderating effects of in-person and online social support on the associations between caregiver outcomes and burden in family caregivers of individuals with SCI.

## 2. Materials and Methods

### 2.1. Design and Procedure

This study used a cross-sectional correlational study design to assess the strengths of the associations between the independent, dependent, and moderating variables [20]. Participants completed a series of measures at one time point. Family caregivers were eligible to participate in this study if they (1) were residents of Canada or the United States; (2) understood and spoke English; and (3) self-identified as the caregiver of an individual with SCI who currently lives in the community. Family caregivers of individuals with SCI who were palliative were excluded. The University of British Columbia Research Ethics Board (H20-01461) approved the study in July 2020. Participants were recruited via social media posts (e.g., Facebook), online promotion from the Wives and Girlfriends of SCI, and advertisements posted on SCI websites. Finally, participants were recruited via word of mouth, emails from the affiliated health research institutes, and printed posters. Advertisements included a link to a survey that contained a description of the study and the eligibility questions. Eligible individuals were asked to review and sign the consent form and complete the survey. Participants could enter a prize draw to win one of two CAD 100 e-gift cards.

### 2.2. Materials

#### 2.2.1. Demographic Questionnaire

The study included a researcher-generated self-administered online survey to collect information on the participants’ demographic information, such as age, sex, and location, as well as information about their relationship with the individual with SCI and their caregiving responsibilities.

#### 2.2.2. Perceived Support

The perceived social support scale [21] was used to measure the impact of social support on caregivers. The scale includes nine items that measure self-help support, information exchange, and emotional support [21]. This scale includes items such as “I can talk over my feelings about caregiving with others who have similar values,” and “Others I know have helped me deal with frustrations I have as a result of being a caregiver.” In the study, two versions of the scale were used, i.e., in-person and online support versions. Online support measured the support participants receive from those they met through social media, not including immediate family, close friends, or health care providers. The questions in both versions were similar, with slight differences. For example, “Others I know in-person have given me information about Spinal Cord Injury” was used to measure in-person support and “Others on social media have given me information about Spinal Cord Injury” was used to measure online support. Participants ranked their responses on the five-point Likert scale ranging from not at all (0) to always (4). A higher score indicates greater support. The Cronbach’s alpha scores in the current study were α = 0.92 (in-person version) and α = 0.93 (online version).

#### 2.2.3. Caregiver Burden

The Caregiver Burden Inventory in SCI (CBI-SCI) is a modified version of the CBI and consists of 24 items. The scale has five subscales measuring time-dependent, physical, developmental, social, and emotional burdens [22]. Participants can score each item on the Likert scale ranging from not descriptive at all (1) to strongly descriptive (5); a higher score indicates a higher burden. In a previous study, all five subscales had a Cronbach’s alpha ranging from α = 0.76 to α = 0.91 [22]. In this study, the total scale had a Cronbach’s alpha of α = 0.91.

#### 2.2.4. Relationship Closeness

The Relationship Closeness Scale [23] was used to measure relationship closeness. This scale has six items and asks participants to consider their relationship with their family member or partner with SCI and choose the most applicable response on a four-point scale from strongly disagree (1) to strongly agree (4). The higher the participant scored, the closer the relationship [24]. The Cronbach’s alpha for this measure in this study was 0.69.

#### 2.2.5. Caregiving Competence

To measure caregivers’ self-appraisal of their efficacy at providing care, the Caregiving Competence Scale [25] was used. Items, including “how much do you believe that you’ve learned how to deal with a very difficult situation,” “how much do you feel that all in all, you’re a good caregiver,” “how competent do you feel,” and “how self-confident do you feel,” can be answered on a scale from not at all (0) to very much (4). The higher the participants scored, the more competent they felt as caregivers [25]. In this study, the Cronbach’s alpha score of this scale was 0.73.

#### 2.2.6. Distress

To measure distress, the Depression, Anxiety, Stress Scale with 21 items [26] was used. The depression subscale includes statements regarding low mood and self-esteem; the anxiety subscale measures physiological arousal and perceived panic; and the stress subscale measures tension and irritability. Participants were asked to rate each statement, such as “I found myself getting upset by quite trivial things” or “I tended to over-react to situations,” on a scale ranging from never (0) to almost always (5) based on how much the statement applied to them over the past week. The scale had a Cronbach’s alpha score of 0.93 [26]. The Cronbach’s alpha in our study was 0.89.

### 2.3. Analysis

Descriptive analyses were used to analyze the demographic data. Pearson product-moment correlations were used to investigate the associations between the variables in the moderation analyses. We reviewed the associations among independent variables, moderators, and dependent variables to detect variables with high correlations. This step was important to detect any multicollinearity concerns before conducting the moderation analyses. Any correlations higher than 0.70 among the variables in the model could be a concern for multicollinearity [27], which was not the case in this study. We used Model 2 in the PROCESS computational tool [28], which allowed us to simultaneously investigate the moderating effects of two moderators (online support and in-person support) in each model. The moderated analyses aimed to examine whether the association between the independent variable (i.e., distress, caregiving competence, relationship quality) and the dependent variable (i.e., subscales of burden) could be moderated by our moderators (i.e., online support and in-person support). Variables in the model were centred using a mean-centred feature in the PROCESS. While the multicollinearity was not a concern in this study, mean centring has been used as a recommended method for addressing concerns related to multicollinearity [29]. We performed 15 moderation analyses. When moderation analyses revealed a statistically significant moderator (*p* < 0.05), we performed slope analyses to investigate the nature of the interaction and the level (±1 SD of the moderator’s mean) at which the interaction is statistically significant.

## 3. Results

### 3.1. Demographic Information

The sample consisted of 115 family caregivers. Most were females (96.5%), between 25 and 34 years old (33.9%), and spouses or partners of an individual with SCI (95.7%). Results indicated that 37 participants (32.3%) were caregivers for more than five years, and almost all (93.9%) lived with the individual with SCI. Additional details can be found in [18]. Section A.1 provides detailed information on demographic characteristics.

### 3.2. Correlations

The results of Pearson correlations that assessed the correlations among the independent variables (i.e., relationship quality, caregiver competence, distress), moderating variables (i.e., in-person and online social support), and dependent variables (i.e., time-dependent, developmental, physical, emotional, and social burden) show that in-person support and online support were not correlated (*p* = 0.29). The associations between in-person support and burden, distress, and relationship quality were non-significant. Online support had a positive and statistically significant association with burden (r = 0.27, *p* = 0.007) and distress (r = 0.259, *p* = 0.010). Online support did not have an association with relationship quality. There was a positive association between distress and burden (r = 0.667, *p* < 0.001) and a negative association between relationship quality and burden (r = −0.41, *p* < 0.001). Section A.2 provides detailed information on the Pearson correlation results.

### 3.3. Moderating Role of In-Person and Online Supports

Below, we presented the results that had at least one significant moderator. Table 1, Table 2 and Table 3 present the complete results of the moderation analyses.

#### 3.3.1. In-Person and Online Supports Moderate the Association Between Relationship Quality and Burden

Physical burden as the dependent variable. Only the main effect of online support on physical burden was significant (B = 0.20, *p* = 0.042). The moderating effect of online support on the associations between relationship quality and physical burden was also statistically significant (B = −0.58; *p* = 0.019). Slope analyses indicated that the negative association between relationship quality and the physical burden was more strongly related to higher levels of online support (*p* = 0.013) and not to lower levels of online support (*p* = 0.75). This means that stronger relationship quality is associated with lower physical burden when caregivers reported higher levels of online support. Figure 1 presents the ±SD slopes.

Emotional burden as the dependent variable. The analyses revealed statistically significant main effects of relationship quality (B = −0.58, *p* < 0.001) and in-person support (B = 0.101, *p* = 0.037) on emotional burden. In addition, the moderating effect of in-person support on the link between relationship quality and emotional burden was significant (B = −0.52; *p* < 0.001). Slope analyses showed that the negative link between relationship quality and emotional burden is significant when caregivers reported higher levels of in-person support (*p* < 0.001), meaning that while better relationship quality is linked to lower emotional burden, this link is statistically significant when caregivers reported higher in-person support. Figure 2 presents the ±SD slopes.

**Table 1 ijerph-22-01075-t001:** Moderating Effects of Online and In-Person Support on the Association Between Relationship Quality and Caregiving Burden.

Dependent	Independent Variables	Unstandardized B Coefficient	Se	t	*p*	LLCI	ULCI
Burden Time-Dependence ^1^	Relationship quality	−0.48	0.30	−1.59	0.11	−1.09	0.12
	Online support	0.14	0.10	1.34	0.18	−0.07	0.36
	Online support × Relationship quality	−0.16	0.27	−0.61	0.53	−0.71	0.37
	In-person support	0.05	0.11	0.49	0.62	−0.17	0.28
	In-person support × Relationship quality	−0.24	0.25	−0.93	0.35	−0.75	0.27
Burden Developmental ^2^	Relationship quality	−0.95	0.25	−3.77	0.00	−1.46	−0.45
	Online support	0.30	0.09	3.38	0.00	0.12	0.48
	Online support × Relationship quality	−0.41	0.22	−1.81	0.07	−0.86	0.04
	In-person support	−0.04	0.09	−0.46	0.64	−0.23	0.14
	In-person support × Relationship quality	−0.23	0.21	−1.09	0.27	−0.66	0.19
Burden Physical ^3^	Relationship quality	−0.510	0.27	−1.85	0.06	−1.06	0.03
	Online support	0.203	0.09	2.06	0.04	0.00	0.39
	Online support × Relationship quality	−0.58	0.24	−2.38	0.01	−1.08	−0.09
	In-person support	−0.03	0.10	0.31	0.75	−0.17	0.23
	In-person support × Relationship quality	0.05	0.23	0.25	0.80	−0.40	0.52
Burden Social ^4^	Relationship quality	−0.80	0.10	−3.22	0.00	−1.30	−0.30
	Online support	0.27	0.08	3.07	0.00	0.09	0.45
	Online support × Relationship quality	−0.27	0.22	−1.21	0.22	−0.72	0.17
	In-person support	−0.03	0.09	−0.41	0.68	−0.22	0.14
	In-person support × Relationship quality	0.04	0.21	−0.23	0.81	−0.37	0.47
Burden Emotionl ^5^	Relationship quality	−0.58	0.12	−4.56	0.00	−0.84	−0.33
	Online support	0.04	0.04	1.07	0.28	−0.04	−0.14
	Online support × Relationship quality	−0.05	0.11	−0.45	0.64	−0.28	0.17
	In-person support	0.10	0.04	2.11	0.03	0.00	0.19
	In-person support × Relationship quality	−0.52	0.10	−4.85	0.00	−0.74	−0.31

Model summary for each dependent variable: ^1^. R = 0.25, R^2^ = 0.06, F = 1.23, *p* = 0.30; ^2^. R = 0.53, R^2^ = 0.29, F = 7.16, *p* = < 0.001; ^3^. R = 0.39, R^2^ = 0.15, F = 3.19, *p* = 0.01; ^4^. R = 0.48, R^2^ = 0.23, F = 5.29, *p* = 0.0003; ^5^. R = 0.58, R^2^ = 0.33, F = 8.96, *p* < 0.001.

#### 3.3.2. In-Person and Online Supports Moderate the Relationship Between Caregiver Competence and Burden

Physical burden as the dependent variable. The main effects of competence (B = 0.042; *p* = 0.87) and in-person support (B = −0.01; *p* = 0.87) on burden were not significant. However, the main effect of online support on physical burden was significant (B = 0.30, *p* = 0.009). Only the moderating effect of online support on physical burden was statistically significant (B = −0.73; *p* = 0.013). The slope analyses showed that the relationship between competence and the physical burden was significant if caregivers reported high levels of online support (*p* = 0.037) and not when caregivers reported lower levels of online support (*p* = 0.60). In other words, higher competence was associated with lower physical burden only when online support was high. This suggests that online support may strengthen the positive effects of caregiver competence in reducing physical burden. Figure 3 demonstrates the SD slopes.

Emotional burden as an outcome. Only the main effect of competence on emotional burden was significant (B = −0.34, *p* = 0.012). Moreover, in-person support had a statistically significant moderating effect on the link between competence and emotional burden (B = −0.35, *p* = 0.001). The slope analyses revealed that the relationship between competence and emotional burden is significant if caregivers perceived high levels of in-person support (*p* = 0.007) and not when they perceived lower levels of in-person support (*p* = 0.81). This indicates that higher levels of competence were linked to lower emotional burden only when in-person support was high, suggesting that in-person support enhances the emotional benefits of caregiver competence. Figure 4 demonstrates the ±SD slopes.

**Table 2 ijerph-22-01075-t002:** Moderating Effects of Online and In-Person Support on the Association Between Caregiver Competence and Caregiving Burden.

Dependent Variable	Independent Variables	Unstandardized B Coefficient	Se	t	*p*	LLCI	ULCI
Burden Time-Dependence ^1^	Competence	0.23	0.29	0.80	0.42	−0.35	0.83
	Online support	0.20	0.12	1.63	0.10	−0.04	0.45
	Online support × Competence	−0.23	0.31	−0.74	0.46	−0.86	0.39
	In-person support	0.003	0.11	0.03	0.97	−0.22	0.23
	In-person support × Competence	−0.14	0.23	−0.63	0.52	−0.61	0.31
Burden Developmental ^2^	Competence	−0.38	0.26	−1.43	0.15	−0.91	0.14
	Online support	0.31	0.11	2.75	0.007	0.086	0.53
	Online support × Competence	−0.19	0.28	−0.85	0.49	−0.76	0.37
	In-person support	−0.13	0.10	−1.36	0.17	−0.34	0.06
	In-person support × Competence	−0.20	0.21	−0.95	0.34	−0.62	0.21
Burden Physical ^3^	Competence	0.04	0.27	0.15	0.87	−0.49	0.58
	Online support	0.30	0.11	2.67	0.009	0.07	0.53
	Online support × Competence	−0.73	0.29	−2.52	0.01	−1.31	−0.15
	In-person support	−0.016	0.10	−0.15	0.87	−0.22	0.19
	In-person support × Competence	−0.14	0.21	−0.69	0.49	−0.57	0.28
Burden Social ^4^	Competence	−0.004	0.25	0.015	0.98	−0.51	0.51
	Online support	0.27	0.11	2.53	0.01	0.05	0.49
	Online support × Competence	0.01	0.27	0.06	0.94	−0.53	0.57
	In-person support	−0.09	0.09	−0.97	0.33	−0.29	0.10
	In-person support × Competence	0.14	0.20	0.69	0.48	−0.26	0.55
Burden Emotionl ^5^	Competence	−0.34	0.13	−2.55	0.01	−0.61	−0.07
	Online support	0.02	0.05	0.37	0.71	−0.09	0.13
	Online support × Competence	0.06	0.14	0.45	0.64	−0.22	0.35
	In-person support	0.02	0.05	0.43	0.60	0.08	0.12
	In-person support × Competence	−0.35	0.10	−3.25	0.001	−0.56	−0.13

Model summary for each dependent variable: ^1^. R = 0.18, R^2^ = 0.03, F = 0.59, *p* = 0.70; ^2^. R = 0.38, R^2^ = 0.14, F = 2.94, *p* = 0.017; ^3^. R = 0.31, R^2^ = 0.10, F = 2.00, *p* = 0.086; ^4^. R = 0.31, R^2^ = 0.102, F = 2.02, *p* = 0.08; ^5^. R = 0.43, R^2^ = 0.19, F = 4.11, *p* = 0.002.

#### 3.3.3. In-Person and Online Supports Moderate the Relationship Between Distress and Burden

Social burden as the dependent variable. The main effects of distress (B = 1.48; *p* < 0.001) and online support (B = 0.20; *p* = 0.017) on burden were significant. However, the main effect of in-person support and burden was not statistically significant (B = −0.075; *p* = 0.36). In addition, only the moderating effect of online support on the link between social burden and distress was significant (B = 0.47; *p* = 0.029). Slope analyses showed that the relationship between social burden and distress was significant if caregivers reported high levels of online support (effect = 2.27; *p* < 0.001), and it was not significant when caregivers reported lower levels of online support (effect = 0.71; *p* = 0.10). In summary, higher distress was associated with greater social burden only when online support was high. Figure 5 depicts the ±SD slopes.

Emotional burden as an outcome. The main effect of distress on emotional burden was significant (B = 0.75; *p* < 0.001), and the moderating effect of online support on the link between emotional burden and distress was significant (B = 0.26; *p* = 0.045). The slope analyses showed that the relationship between distress and emotional burden was significant if caregivers reported high levels of online support (*p* < 0.001) and not when caregivers reported lower levels of online support (*p* = 0.60). This indicates that higher distress was associated with greater emotional burden only when online support was high. Alternatively, it is also possible that caregivers experiencing greater distress are more likely to seek out online support, which can be interpreted as a need for help. Figure 6 depicts the ±SD slopes.

**Table 3 ijerph-22-01075-t003:** Moderating Effects of Online and In-Person Support on the Association Between Distress and Caregiving Burden.

Dependent	Independent Variables	Unstandardized B Coefficient	Se	t	*p*	LLCI	ULCI
Burden Time-Dependence ^1^	Distress	1.17	0.33	3.53	0.00	0.51	1.84
	Online support	0.07	0.10	0.72	0.47	−0.13	0.29
	online support × distress	0.31	0.27	1.13	0.26	−0.23	0.86
	In-person support	0.01	0.10	0.17	0.85	−0.19	0.23
	In-person support × distress	0.27	0.31	0.86	0.39	−0.35	0.90
Burden Development ^2^	Distress	1.78	0.25	6.86	0.00	1.26	2.30
	Online support	0.17	0.08	2.10	0.03	0.00	0.34
	Online support × distress	0.28	0.21	1.32	0.19	−0.14	0.71
	In-person support	−0.10	0.08	−1.26	0.20	−0.27	0.06
	In-person support × distress	0.14	0.24	0.56	0.57	−0.35	0.63
Burden Physical ^3^	Distress	1.62	0.28	5.69	0.00	1.05	2.18
	Online support	0.04	0.09	0.50	0.61	−0.13	0.23
	Online support × distress	0.12	0.23	0.54	0.58	−0.34	0.60
	In-person support	0.01	0.09	0.15	0.87	−0.16	0.19
	In-person support × distress	−0.09	0.27	−0.36	0.72	−0.63	0.44
Burden Social ^4^	Distress	1.48	0.25	5.72	0.00	0.96	1.99
	Online support	0.20	0.08	2.42	0.01	0.03	0.37
	Online support × distress	0.47	0.21	2.21	0.02	0.04	0.90
	In-person support	−0.07	0.08	−0.90	0.36	−0.24	0.09
	In-person support × distress	0.17	0.24	0.71	0.47	−0.31	0.66
Burden Emotionl ^5^	Distress	0.75	0.15	4.8	0.00	0.44	1.06
	Online support	0.01	0.05	0.30	0.76	−0.08	0.11
	Online support × distress	0.26	0.13	2.03	0.04	0.00	0.52
	In-person support	0.04	0.05	0.80	0.42	−0.05	0.14
	In-person support × distress	0.28	0.14	1.91	0.05	−0.01	0.57

Model summary for each dependent variable: ^1^. R = 0.40, R^2^ = 0.16, F = 3.31, *p* = 0.0086; ^2^. R = 0.66, R^2^ = 0.43, F = 13.64, *p* < 0.001; ^3^. R = 0.56, R^2^ = 0.31, F = 8.22, *p* < 0.001; ^4^. R = 0.62, R^2^ = 0.38, F = 10.90, *p* < 0.001; ^5^. R = 0.51, R^2^ = 0.26, F = 6.27, *p* < 0.001.

## 4. Discussion

The findings showed that the negative association between relationship quality and physical burden was stronger at higher levels of online support. Similar results were found for caregivers’ competence. We also found that the negative associations between relationship quality, emotional burden, and caregiver competence and emotional burden were stronger at the higher levels of in-person social support. Finally, we found that the associations between distress and social and emotional burden were only statistically significant when family caregivers reported high levels of online support.

Research has shown that caregivers receiving online support have better relationship quality with their care recipient [30,31,32]. In online support groups, caregivers give advice on how to continue leisure activities, such as travelling, which allows caregivers to spend quality time with their care recipient [33]. Moreover, Tixier et al. [34] showed that receiving online support helped caregivers to learn how to communicate better with their care recipients. Better communication improves the caregiver–care recipient relationship and helps them to understand the care recipient’s physical needs, which makes providing care easier and potentially reduces the physical burden [34].

However, other research suggests that social support, receiving support related to one’s well-being through a social network [35], can negatively affect caregivers’ well-being and their relationship quality with the care recipient [36]. In Kreuter et al.’s [37] study, some caregivers expressed their frustration with receiving social support, especially when they perceived that the members of their social networks who were providing the support were interfering with their lives, which could cause conflict between the caregiver and care recipient, reducing their relationship quality. While previous research primarily focused on evaluating the effects of in-person social support, the current study considered both types of receiving social support, i.e., in-person and online. The findings of this study showed that the negative association between the relationship quality and physical burden was stronger at higher levels of online support. This may indicate that receiving support online can benefit family caregivers, as it may not have the same negative consequences as in-person social support, such as interfering in the caregiver–care recipient relationship. Our results also showed that online social support moderated the relationship between caregivers’ competence and physical burden. Caregivers of individuals with Alzheimer’s found helpful tips in online groups, such as examples of good practices to follow, complex situations that may happen, and how to assess pain [34]. Caregivers of individuals with Alzheimer’s who were part of the online groups felt less stressed about caregiving tasks, which left them feeling more competent, and some of the suggestions shared in the group helped them to lighten their physical burden [34].

Furthermore, family caregivers who received in-person support from nurses and home support workers, such as how to administer medication, felt more competent [38]. Having nurses listen to their concerns, work with them to make decisions, and discuss expectations regarding the recovery journey lessened caregivers’ emotional burden as they felt better prepared [38]. Moreover, home support workers build relationships with care recipients, and these relationships can provide the care recipient with companionship and social interaction, improving their well-being. Therefore, the support worker meets some of the care recipients’ social and emotional needs, reducing the caregiver burden [39], which supports our research findings showing that caregivers who receive in-person support display higher levels of competence and less emotional burden. As the current study did not investigate the effects of receiving in-person support from a health care provider versus a family or a friend [34] further studies are needed to evaluate the benefits of various sources of in-person support on the well-being of family caregivers.

Previous studies have shown that in-person social support can act as a buffer, reducing caregiver distress and alleviating burden by fulfilling their needs through providing informational and emotional support [40,41]. However, in contrast to previous studies that mainly investigated in-person support, the current study examined in-person and online social support. The findings of this study showed that the association between distress and social and emotional burden were stronger at higher levels of online support. This highlights the negative impact of online support on caregiver distress and burden. Supporting evidence suggests that caregivers of hospice patients with cancer expressed difficulties in being regularly exposed to other caregivers’ grief in online Facebook groups [11]. These caregivers also expressed hesitation to share information that might upset other caregivers in the group; therefore, they did not always receive the support they needed [11]. These experiences resulted in the caregiver feeling more distressed and increased their burden. For family caregivers already experiencing high levels of distress, the additional effort it takes to engage in the group demotivates them from fully participating [42]. The constant connectivity of social media and information overload overwhelmed some caregivers as they felt obligated to engage in discussions, contributing to their social and emotional burden [11]. It can also be argued that it is not the online activity that affects the association between distress and burden, but that caregivers who are experiencing greater distress have an increased likelihood of using online sources. Specifically, factors such as cost and accessibility may encourage the use of online options [27]; hence, when caregivers are experiencing heightened distress and navigating barriers, they are more likely to resort to online social support. Furthermore, distressed individuals use online support for a second opinion or reassurance [43]. Therefore, more distressed caregivers may increase their engagement with online activities. Future longitudinal studies should investigate the causal relationship between distress and online social activities.

Despite our study’s novel contributions of examining the effects of in-person and online social support on the well-being of family caregivers, the current study’s findings should be reviewed with several limitations. The generalizability of the sample is limited as most of our participants were female and partners of care recipients, although this reflects the population, as most caregivers are female [44]. The participants were mainly recruited online, excluding caregivers who did not use social media. However, recruiting caregivers online allowed us to target caregivers who already access online support and explore how their experiences in receiving online support and in-person support differ. In our study, the participants did not clarify if in-person support was from friends and family or healthcare providers, making it challenging to compare participants’ experiences, as research has shown that caregivers perceive social support from family and friends differently from how they perceive support from healthcare providers [41,45]. The cross-sectional and self-report nature of this study also prevented us from drawing causal conclusions. Finally, we could not assess the associations between various sources of online and in-person support (e.g., friends, family, healthcare providers) and caregivers’ well-being. Therefore, this study cannot provide information about when online or in-person support may have higher benefits for family caregivers.

## 5. Conclusions

The findings of this study improve our understanding of the potential ways in which in-person and online social support are associated with caregivers’ well-being. In addition, this study disaggregated the burden into time, developmental, emotional, social, and physical burden, showing the unique pathways through which online or in-person support can be beneficial. This study showed a unique association between distress and burden at high levels of online support, and, while this finding requires additional exploration, it can be used to develop targeted support strategies for caregivers in distress. Considering the increase in use of online social networks, providing safe, evidence-based online resources and facilitating online engagement in a safe and reliable environment for family caregivers is essential. Furthermore, the availability and accessibility of in-person and online peer support to address caregivers’ emotional, social, and informational needs should be considered. Future studies should focus on identifying the elements of in-person and online support that positively and negatively contribute to the well-being of family caregivers. Future research should also aim to develop online and in-person communities where family caregivers receive support without facing factors that add to their distress.

## Figures and Tables

**Figure 1 ijerph-22-01075-f001:**
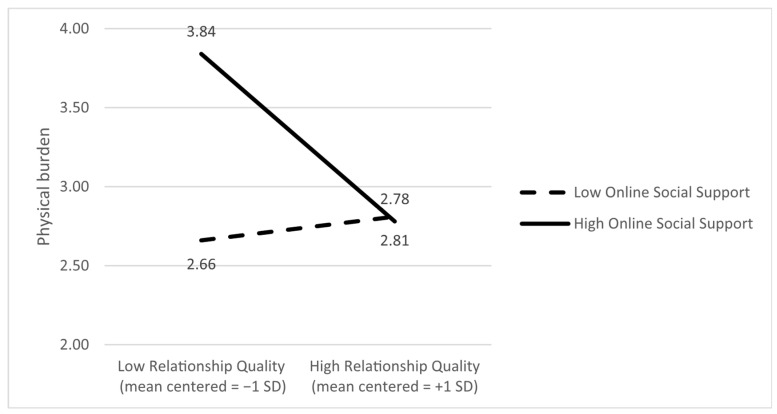
Online social support moderates the association between relationship quality and physical burden; that is, the negative association between relationship quality and physical burden is stronger at higher levels of online support.

**Figure 2 ijerph-22-01075-f002:**
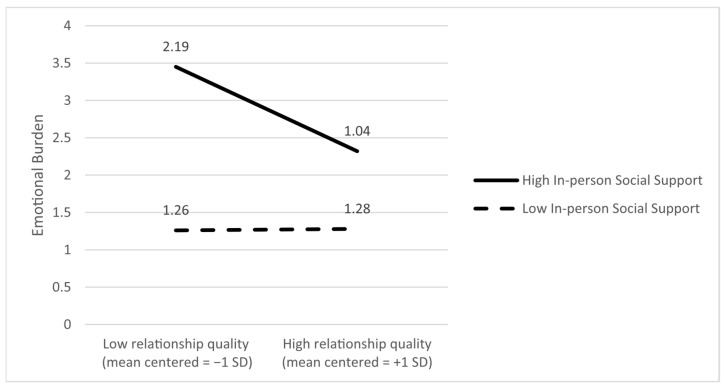
In-person social support moderates the association between relationship quality and emotional burden; that is, the negative association between relationship quality and emotional burden is stronger at higher levels of in-person support.

**Figure 3 ijerph-22-01075-f003:**
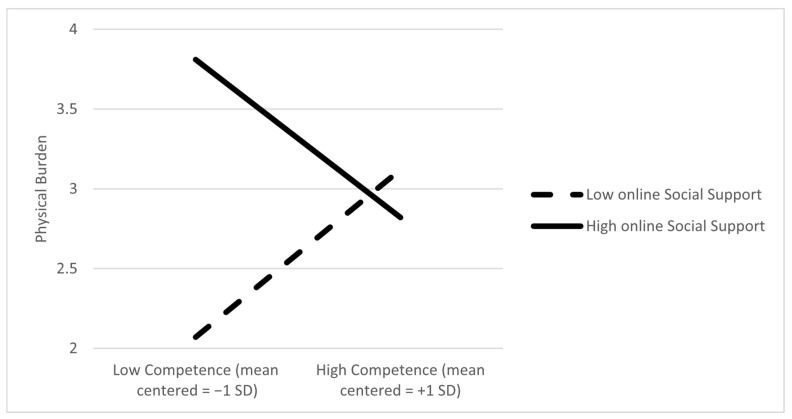
Online social support moderates the association between caregiver competence and physical burden; that is, higher competence was associated with lower physical burden only when online support was high.

**Figure 4 ijerph-22-01075-f004:**
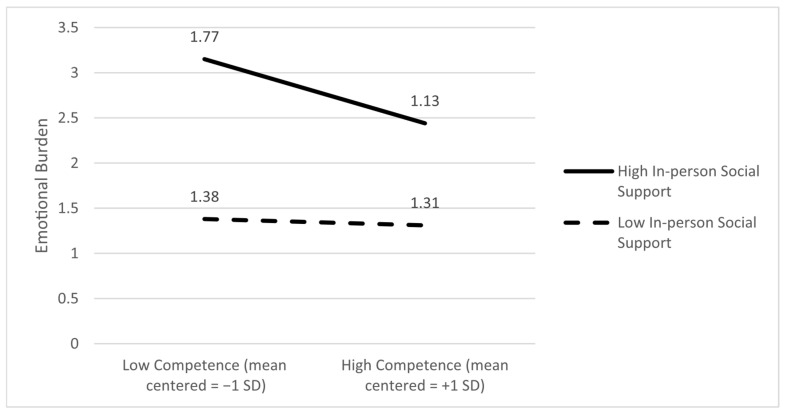
In-person social support moderates the association between caregiver competence and emotional burden; that is, higher competence was associated with lower emotional burden only when in-person support was high.

**Figure 5 ijerph-22-01075-f005:**
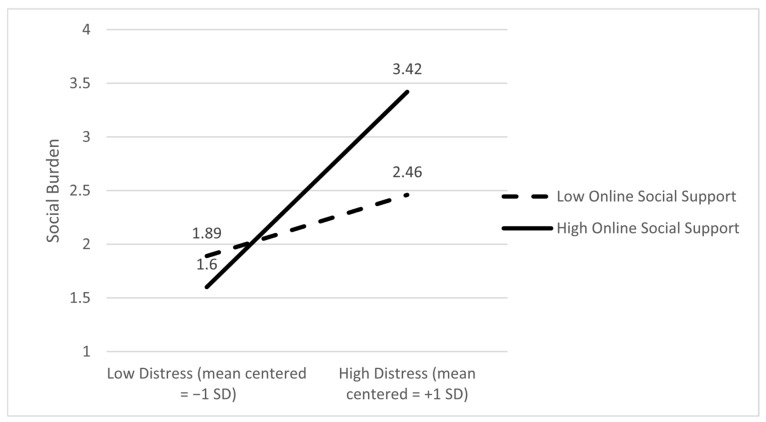
Online social support moderates the association between caregiver distress and social burden; that is, higher distress was associated with higher social burden only when online support was high.

**Figure 6 ijerph-22-01075-f006:**
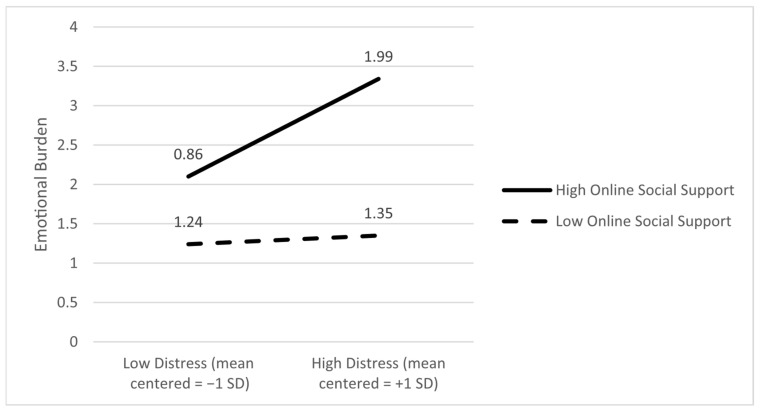
Online social support moderates the association between caregiver distress and emotional burden; that is, higher distress was associated with higher emotional burden only when online support was high.

## Data Availability

The data presented in this study are available on request from the corresponding author.

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
