# Peer review of "Well-Being of Family Caregivers of Individuals with Spinal Cord Injury: The Moderating Effects of Online Versus In-Person Social Support"

_ijerph, 2025, doi:10.3390/ijerph22071075_

Round 1

Reviewer 1 Report

Comments and Suggestions for Authors

This descriptive study explores the intricate dimensions of well-being experienced by family caregivers of individuals with spinal cord injuries who require substantial assistance. The paper is generally well-written. However, after careful reading, various sections of the study reveal a blending of moderation roles and impact interpretations, which necessitates a clearer presentation of survey questions to enhance the comprehension of the results. While both the analytical methods and the study results hold significance, the uniqueness of this research compared to existing literature lacks sufficient clarity. The title, objectives, results, interpretations, and discussion should all be aligned and consistent with each other. Additionally, it is crucial to acknowledge and discuss the other limitations of the study, along with the measures taken to address them, to provide a comprehensive understanding of the validity of the study. Study implications for the policy and practice should be noted. Careful attention should also be given to reviewing the abstract and conclusions to ensure they accurately reflect the research findings and their implications. Please see the other details of the comments and suggestions in the attachment. Good luck!

Author Response

Main Comments

Comment. “Various sections of the study reveal a blending of moderation roles and impact interpretations, which necessitates a clearer presentation of survey questions to enhance the comprehension of the results.”

Response. We modified the manuscript's wording when discussing this study's objective and findings to be more aligned with what we measured, i.e., effects rather than impacts. As we do not have the reprint permission of the scales, we cannot provide the complete measures; however, we added additional examples of the items to help understand the results.

Comment. “While both the analytical methods and the study results hold significance, the uniqueness of this research compared to existing literature lacks sufficient clarity.”

Response. We added more information about the unique aspects of this study and our findings on lines 91-92 and lines 439-445.

Comment. “The title, objectives, results, interpretations, and discussion should all be aligned and consistent with each other.”

Response. We modified the wording used in the title, objectives, results, and discussion to avoid using causal language such as “buffer” and to avoid wordings that imply broader impacts.

Comment. “Additionally, it is crucial to acknowledge and discuss the other limitations of the study, along with the measures taken to address them, to provide a comprehensive understanding of the validity of the study. Study implications for the policy and practice should be noted.

Response. Thank you for your comment. We provided more information about the study’s limitations and implications on lines 432-436 and lines 445-449.

Comment. “Careful attention should also be given to reviewing the abstract and conclusions to ensure they accurately reflect the research findings and their implications.”

Response. We modified and revised the abstract and the discussion to ensure they align with the results. For example, please see lines 31-34 and lines 410-419.

Comments Added to the Abstract

Comment. “What indicators of negative impacts are in the focus of your study? To avoid confusion, define negative impacts and moderating effects carefully. Please, review the discussion of your study in order to link it with the objective.”

Response. We briefly defined the negative impacts, please see line 14.

Comment. “The methods section lacks the info on the study design.”

Response. We added “cross-sectional correlational design” to define our study design. Please see line 16.

Comment. “In medicine and public health, there is a difference between the effect and impact, and diverse methods and indicators are used for their assessment. What are the indicators of moderating impact (paper title) and effect (study objective) in your study? Please review the full text to link it with the study objective.”

Response. Thank you for pointing this out. To address this comment, we modified the title and the other sections of the text and used “effect” when discussing the immediate outcomes of this study.

Comment. “Full term, and acronym”

Response. We replaced SCI with “Spinal Cord Injury”.

Comment. “Please, provide the evidence.”

Response. We provided the results of the moderation analyses in the results section. Please see lines 23-28.

Comment. “Please, explain what type of analysis?”

Response. We replaced “analyses” with “moderation analyses”.

Comment. “This is unclear. Please, explain what was compared and how.”

Response. We tried to explain the result better by adding the following explanation on lines 30-33.

“While online support is usually considered beneficial, when the distress is high, greater online engagement can contribute to higher levels of burden, or it is also possible that caregivers who are more distressed engage more with online media to receive support.”

Comments Added to the Introduction

Comment. “Please, explain what was compared and who, and how the comparison was done. Though the sentence seems logical, it lacks clarity: e.g., was the comparison made between matched adults with and without disease, self-assessed needs for daily activities support, etc.”

Response. The population was “aging persons in the general population”; however, our access to the rest of this report was limited, so we removed this sentence.

Comments Added to the Materials and Methods

Comment: “The study design needs to be explained in detail.”

Response. We explained the study design as follows on lines 97-99:

“This study used a cross-sectional correlational study design to assess the strengths of the associations between the independent, dependent, and moderating variables [20]. Participants completed a series of measures at one time point.”

Comment. “Date of approval? Which Institute? What exactly was approved?”

Response. We added the requested information to lines 103-104.

Comments. “What was the type of the instrument? And the collection method is unclear.”

Response. We provided the requested information, i.e. “self-administered online survey” on line 114.

Comment. “Please, provide it in the supplemental file, if possible. This could help us understand the questions used to measure the scale items.”

Response. As we do not have permission to reprint this scale, we cannot provide the complete measure; however, we added two additional examples of the items. Please see lines 121-123.

Comment. “Please, provide it in the supplemental file, if possible. This could help us understand the questions used to measure the scale items.”

Response. We added the items assessing the caregiver’s competence on lines 151-154.

Comment. “Please adhere to the reporting standards for the statistical parameter in journals.”

Response. We edited the text and followed the statistical parameters of the journal.

Comment. “Please, provide it in the supplemental file, if possible. This could help us understand the questions used to measure the scale items.”

Response. As we do not have permission to reprint this scale, we cannot provide the complete measure; however, we added two additional examples of the items. Please see lines 161-163.

Comments Added to the Results

Comment. “Where is the statistical output of this analysis? What items were measured?”

Response. We added information on the variables being included in the Pearson correlation analysis. In addition, the result of the Pearson correlation can be found in Appendix A.2. We added this information to lines 192-193 and lines 206-207.

Comment. “The title needs to be corrected and avoid using word results.” Use “Independent variables. “Please, add the full term of the coefficient. And please, what is the legend about?

Response. In all three tables, we modified the titles, used “independent variables”, and used the Unstandardized B Coefficient.

Comment. “Please, provide a more appropriate interpretation.”

Response. We modified the explanation by indicating, “This suggests that online support may strengthen the positive effects of caregiver competence in reducing physical burden.”

Please see lines 264-266.

Comment. “Hard to read x-axis. Please, present the scale.”

Response. Values on the x-axis represent mean-centred scores of the independent variable plotted at -1 SD and +1 SD. We added this information to all the figures.

Comment. “or, these persons were seeking more online support.”

Response. Thank you for your comment. We agree with the reviewer that it is also possible that people with higher distress would also engage in a higher level of online activities. As this is a cross-sectional study, we cannot be sure about the directionality of the associations. We also modified the interpretation to raise the point mentioned by the reviewer. Please see lines 319-320.

Comments Added to the Discussion

Comment. “The study findings are from a convenient small sample of unmatched study participants, please, correct the wording.”

Response. We removed the word “buffer” and avoided a strong causal language. The sentence has changed to “The findings showed that the negative association between relationship quality and physical burden was stronger at higher levels of online support.” Please see lines 344-345.

Comment. “More likely is the reverse situation. Please, check how the questions were framed.”

Response. We removed this sentence from this paragraph to ensure the primary focus of this paragraph is on providing a summary of the results. We addressed the point raised in this comment on lines 410-419.

Comment. “Need to define the term” social support.

Response. We defined “Social support”, please see lines 359-360-353.

Comment. “The social support concept is more than having a person at home. Please, correct.”

Response. We corrected the wording in this language to avoid presenting the support received at home as an equivalent of all types of social support.

Comment. “unclear”

Response. We rewrote this section to make it more straightforward. Here, we also emphasize the differences between this study and previous studies. Please see lines 365-371.

“While previous research primarily focused on evaluating the effects of in-person social support, the current study considered both types of receiving social support, i.e. in-person and online. The findings of this study showed that the negative association between the relationship quality and physical burden was stronger at higher levels of online support. This may indicate that receiving support online can benefit family caregivers, as it may not have the same negative consequences as in-person social support, such as interfering in the caregiver and care-recipient relationship.”

Comment. “Which group?” of caregivers.

Response. We defined the group, i.e. Caregivers of individuals with Alzheimer's

Comment. “What support?”

Response. We change “Support” to “online social support.”

Comment. “What is a novelty of your study in relation to the previous research? Please, add that information.”

Response. We provided an explanation on lines 396-401 to explain the additional element of this study.

“However, in contrast to previous studies that mainly investigated in-person support, the current study examined in-person and online social support. The findings of this study showed that the association between distress and social and emotional burden were stronger at higher levels of online support. This highlights the negative impact of online support on caregiver distress and burden.”

Comment. “What was that study about?”

Response. We added additional information about this study. Please see lines 401-403.

Comment. “Please, add an explanation on how you overcame other biases in the study.”

Response. Thank you for your comment. We discussed some of the other biases and limitations we encountered, such as this study's self-reported and cross-sectional nature. We also acknowledge the limitations we can face in interpreting the data based on these limitations. We also modified the wording in the results to avoid causal language. For example, please see lines 432-436.

Comments Added to the Discussion

Comment. “Results in tables, figures, and study methods are not supporting this conclusion.”

Response. We removed this conclusion.

Reviewer 2 Report

Comments and Suggestions for Authors

Dear Authors,

Thank you for the opportunity to review your manuscript. Your study addresses a timely and relevant issue regarding caregiver burden in the context of spinal cord injury (SCI) and offers novel insights into the moderating roles of in-person and online social support. The manuscript is generally well-structured, methodologically sound, and clearly presented. However, there are a few areas that could benefit from clarification or enhancement to strengthen the paper further. My detailed comments are below.

1 - Strengths:

  • The research question is clearly defined, and the study aims are appropriately stated.
  • The study design is suitable for addressing the objectives, and ethical procedures are followed.
  • The paper is logically structured with an adequate review of relevant literature.
  • The statistical analysis (moderation using Hayes’ PROCESS) is well-justified and correctly interpreted.
  • The discussion provides a balanced interpretation of the findings, including the dual role of online support.

2 - Areas for Improvement:

2.1 Discussion Redundancy and Length: Several points are repeated across subsections of the discussion, especially regarding the benefits and drawbacks of online support. My suggestion is to condense and integrate similar findings to improve clarity and reduce length. This will make the discussion more concise and impactful.

2.2 Clarity in Describing Sources of In-person Support: The study does not differentiate whether in-person support is from healthcare professionals, friends, or family. If possible, clarify in the discussion that this ambiguity limits interpretability, and recommend future studies to distinguish sources of in-person support.

2.3 Figures and Captions: While the figures are informative, some lack detailed captions. Please expand figure captions to ensure they are understandable independently of the main text. Describe the nature of moderation and what each gradient represents.

2.4 Language and Grammar: The manuscript is generally well-written; however, a few phrases could be improved for fluency and professionalism (e.g., “In another word” should be “In other words”). A minor language edit throughout the manuscript could improve flow and readability.

2.5 Specialist Statistician Consultation: While statistical methods are appropriate and well-reported, some readers may benefit from further clarification on assumptions or multicollinearity in moderation analysis. You may consider a brief mention of testing these assumptions to reassure readers of the robustness of the models.

Conclusions: You have made a valuable contribution to the field of caregiver research, particularly regarding the nuanced roles of online and in-person support. With minor revisions, particularly in condensing the discussion, enhancing figure descriptions, and clarifying some details in the methodology and language, the manuscript can be ready for publication.

Comments on the Quality of English Language

The manuscript is generally well-written; however, a few phrases could be improved for fluency and professionalism (e.g., “In another word” should be “In other words”). A minor language edit throughout the manuscript could improve flow and readability.

Author Response

Comment. Discussion Redundancy and Length: Several points are repeated across subsections of the discussion, especially regarding the benefits and drawbacks of online support. My suggestion is to condense and integrate similar findings to improve clarity and reduce length. This will make the discussion more concise and impactful.

Response. Thank you for your comments. We reviewed the discussion and made it more concise.

Comment. Clarity in Describing Sources of In-person Support: The study does not differentiate whether in-person support is from healthcare professionals, friends, or family. If possible, clarify in the discussion that this ambiguity limits interpretability, and recommend future studies to distinguish sources of in-person support.

Response. Thank you for raising this point. We provided examples of sources of support, and as this study did not measure these sources separately, we discussed this as a limitation. Please see lines 433-436.

Comment. Figures and Captions: While the figures are informative, some lack detailed captions. Please expand figure captions to ensure they are understandable independently of the main text. Describe the nature of moderation and what each gradient represents.

Response. We modified the captions for all six figures to help the reader understand the result without relying on the text.

Comment. Language and Grammar: The manuscript is generally well-written; however, a few phrases could be improved for fluency and professionalism (e.g., “In another word” should be “In other words”). A minor language edit throughout the manuscript could improve flow and readability.

Response. We edited the text thoroughly.

Comment. Specialist Statistician Consultation: While statistical methods are appropriate and well-reported, some readers may benefit from further clarification on assumptions or multicollinearity in moderation analysis. You may consider a brief mention of testing these assumptions to reassure readers of the robustness of the models.    

Response. Thank you for this helpful comment. To assess the concerns related to multicollinearity, we reviewed the associations among the variables in the model. As the associations among the variables entered in each model were not higher than 0.70, we did not identify any concerns related to multicollinearity. Furthermore, we addressed potential multicollinearity in the moderation analyses by mean-centring all predictor and moderator variables before creating interaction terms—a standard approach that helps reduce non-essential multicollinearity. We provided explanations of the steps we took to assess multicollinearity concerns and address any potential problems on lines 169-173 and lines 179-182.